# Bivalve monitoring over French coasts: multi-decadal records of carbon and nitrogen elemental and isotopic ratios ($\delta^{13}$C, $\delta^{15}$N and C:N) as ecological indicators of global change

Camilla Liénart[1], Alan Fournioux[1], Andrius Garbaras[2], Hugues Blanchet[1], Nicolas Briant[3], Stanislas F. Dubois[4], Aline Gangnery[3], Anne Grouhel Pellouin[3], Pauline Le Monier[3], Arnaud Lheureux[5], Xavier de Montaudouin[1], Nicolas Savoye[1]

[1]Université de Bordeaux, CNRS UMR 5805, Bordeaux INP, EPOC, Pessac, 33600, France
[2]Center for Physical Sciences and Technology, FTMC, Vilnius, 10257, Lithuania
[3]Ifremer Centre Atlantique, CCEM, Nantes, 44000, France
[4]Ifremer, DYNECO, Plouzané, 29280, France
[5]MNHN, CNRS UMR 8067, SU, IRD 207, UCN, UA, BOREA, CP53, F-75005 Paris, France

*Correspondence to*: Camilla Liénart (camilla.lienart@u-bordeaux.fr)

**Abstract**

Recent changes in climate and environment, influenced by both global and local factors, have had profound impacts on coastal ecosystem functioning and trajectories. By examining archived samples from ongoing ecological monitoring efforts, particularly focusing on bivalves like mussels and oysters, we gain a valuable long-term perspective on how ecosystems are responding at various scales. We conducted analyses on carbon and nitrogen content (C, N) and elemental and isotopic ratios (C:N, $\delta^{13}$C, $\delta^{15}$N) of mussel and oyster soft tissues collected annually at 33 sites along the French coast from 1981 to 2021. This extensive dataset (https://doi.org/10.17882/100583, Liénart et al., 2024a) offers a comprehensive view spanning multiple decades and ecosystems, allowing to track how coastal ecosystems and marine species record changing climate, physical-chemical environments and organic matter cycles. Additionally, these data are crucial for establishing isotope baselines for studying food webs. Ultimately, this data set provide valuable information for more effective ecosystem conservation and management strategies in our rapidly changing world.

## 1 Introduction

Over the past two decades, rapid and sometimes abrupt changes in climate and environmental conditions have significantly impacted ecosystems functioning with human activities widely acknowledge as a primary factor driving and modulating these changes (Cloern et al., 2016). At the land-ocean interface, coastal and estuarine environments are particularly vulnerable to the climate change, alongside with local human-induced pressures such as pollution and habitat destruction (Harley et al, 2006; Lotze et al., 2006). In such dynamic environments, pressures vary simultaneously, with different directions and magnitudes, over different time (short vs long-term) and spatial (global vs local) scales (Cabral et al., 2019). Hence, a fundamental challenge persists in elucidating how the various facets of global change affect coastal ecosystems responses at local and regional scales, and consequently the services they provide to human societies. Recurrent ecological monitoring programs conducted at large spatial scales are valuable to assess ecosystem status and the pressures affecting important properties (e.g., biodiversity, carbon and nutrient cycling), and to detect changes over multiple decades (Hofmann et al., 2013; Sukhotin and Berger, 2013). Yet, only a few ecological indicators are both integrative (i.e. measured in long-living species tissues) and sensitive enough (e.g. measured at molecular level such as metallothionein used as a biomarker of metal exposure, Amiard et al., 2006) to various disturbances while also exhibiting predictable responses with a low variability in its response (Dale and Beyeler, 2001; Niemi and McDonald, 2004).

Filter-feeding bivalves include important reef habitat forming species, which promote benthic-pelagic coupling and nutrient recycling (Ray and Fulweiler, 2020). Some of them such as mussels (*Mytilus* spp.) and oysters (*Crassostrea gigas*) represent an economical value through aquaculture and are widely used as bioindicators for climate and environmental change as well as contaminants monitoring (Kanduč et al., 2018; Karlson and Faxneld, 2021; Mazaleyrat et al., 2022; Briand et al., 2023; Chahouri et al., 2023; Liénart et al., 2024b). The use of ecological indicators is the basis of ecosystem monitoring to detect early-warning signals in ecosystem changes or disturbances (Dale and Beyeler, 2001). Indeed, bivalve tissues records the environmental conditions of their sampling area over time. Since the end of the 1970s, the French monitoring network "ROCCH" ("Réseau d'Observation de la Contamination CHimique", coordinated by Ifremer) monitor chemical contamination in the environment along French coastlines by using oysters and mussels as bioindicators (Briant et al., 2018; Chahouri et al., 2023). The ROCCH network focuses on chemical contaminants, but its archived samples represent a large sample bank that can be useful for other ecological purposes, such as understanding long-term changes in ecosystem functioning and responses to global or local pressures.

Measuring elemental and isotopic composition of the main building blocks of life (i.e., carbon and nitrogen; C:N, $\delta^{13}$C, $\delta^{15}$N) is relevant to understand nutrients origin, to determine organisms diet and food webs structure, and can serve as indicators of water quality over space and time (Glibert et al., 2018). The C:N ratio is mostly an indicator of bivalve condition and physiology, reflecting the balance between organisms' requirements and elemental availability in the environment (i.e., ecological stoichiometry sensu Elser et al., 2003; N content increases (thus C:N ratio decreases) as protein content increases whereas C content increases (thus C:N ratio increases) as lipid or carbohydrate content increases). When measured over long-

term in bivalve tissues, such dataset allow scientists to understand ecosystems responses to changing physical-chemical environment and organic matter cycling and to track trends in climate changes and its effect on coastal ecosystems (Liénart et al., 2020, 2024b). We took advantage of the large-scale multi-decadal sample bank of the ROCCH to analyze C and N content and C:N, $\delta^{13}$C and $\delta^{15}$N ratios in mussels (*M. edulis, M. galloprovincialis*) and oysters (*C. gigas*) tissues from a set of 33 stations distributed along the French coasts. In this paper, we present a unique dataset of multi-decadal and multi-ecosystem carbon and nitrogen content and elemental and isotopic ratios from three widespread bivalve species. As long-term isotopic datasets reflect the impact of human activities, such as coastal pollution and habitat alteration, the data derived here can provide valuable information for conservation and management strategies for coastal areas, in helping make informed decisions to mitigate environmental threats and protect vulnerable ecosystems. Ultimately it could provide valuable input for developing predictive models of bivalve physiology (Emmery et al., 2011) or trophic ecology (Marín Leal et al., 2008) explaining ecosystem response to future environmental changes and possibly forecast potential impacts of climate change and human activities on coastal ecosystems. Overall, this long-term dataset of isotopic ratio in suspension-feeders tissues provides insights into ecosystem dynamics and is essential for advancing scientific understanding in the face of ongoing environmental challenges.

## 2 Methods and data

### 2.1 The ROCCH: network and sampling procedure

Over the last four decades, the French national monitoring network for chemical contaminant, the "ROCCH" ("Réseau d'Observation de la Contamination CHimique"), coordinated by Ifremer ("Institut français de recherche pour l'exploitation de la mer", https://littoral.ifremer.fr/Reseaux-de-surveillance/Environnement/ROCCH-Reseau-d-Observation-de-la-Contamination-CHimique-du-littoral) has been annually sampling bivalves as bioindicators of chemical contamination. The number of stations has varied since the initiation of the network, so has the sampling frequency. Approximately 150 stations are nowadays monitored along the French coastlines, which begins in 1979 for historical stations. Sampling took place once a year during winter, currently mid-February, with a tolerance of one tidal cycle before and after the target date, meaning a 6 weeks amplitude spreading from the end of January to the beginning of March on the overall sites of the network. Three different species are targeted, the Pacific oyster *Crassostrea gigas* and the blue mussels *Mytilus edulis* in the English Channel and Atlantic facades, and *M. galloprovincialis* in the Mediterranean facade.

The sampling protocol has been identical since the start of the monitoring and was designed to acquire bivalve samples with consistent and homogeneous characteristics for measuring chemical concentrations of contaminants. Bivalves are collected alive at fixed points (maximum tolerance of 180 m around the selected point) chosen away from known anthropogenic discharges. Bivalves are sourced either from wild beds or from dedicated rearing facilities, ensuring that they remain on site for at least 6 months before being sampled. The selected individuals are adults of the same species and of uniform size (30 to 60 mm long for mussels, 90 to 140 mm long for oysters, i.e., 2 to 3 years old). A minimum of 50 mussels or 10 oysters was

required to constitute a representative pooled sample accounting for inter-individual variability and to get enough material for chemical analysis and long-term storage. Bivalves are first depurated for 18 to 26 hours in decanted seawater collected near the collection site. Next, once extracted from the shell, the whole bodies (i.e., total soft-tissues) of each individual are pooled and drained all together for 30 minutes. The resulting pooled bivalve samples are placed in clean glass containers (washed and baked for 8 hours at 450 °C) covered with calcined aluminum foil, and sealed with a plastic lid and frozen (−20 °C). Frozen samples are sent to the central laboratory in Nantes (France) where the samples are thawed, grounded and homogenized in a stainless steel-bladed blender and freeze-dried. After chemical analysis of a sample aliquot, the containers are indefinitely stored at Nantes IFREMER center, at room temperature, protected from light, under moister-regulation control to prevent moisture pick-up. The original protocol (Grouhel 2023, in French) is available at https://doi.org/10.13155/97878.

**2.2 Study sites and bivalve dataset**

In the present dataset, bivalve samples were selected from 33 stations of the ROCCH network distributed along the three French sea facades: the English Channel (E.C., 10 stations), the Atlantic Ocean (A.O., 17 stations) and the Mediterranean Sea (M.S., 6 stations; Figure 1, Table 1). The stations were selected to span over a wide diversity of geomorphological and environmental conditions encountered by bivalves along the French coastlines. Ecosystems vary from open-sea coasts (rocky or sandy littoral shores), open or semi-enclosed bays and rias, lagoon (shallow water bodies almost closed by narrow landforms), and estuary mouths, ranging from eutrophic to oligotrophic systems, under temperate oceanic or Mediterranean climates, and along gradients of river influence from the main French rivers (Seine, Loire, Garonne, Dordogne, Rhône; Table 2). The complete dataset consist in 1141 bivalve winter samples for time series spanning 11 to 40 years (depending on stations) over the period 1981-2021 (Table 1; https://doi.org/10.17882/100583, Liénart et al., 2024a).

**2.3 Sample analysis and data quality**

Aliquots of 400-700 µg of each dry and grounded archived sample of pooled bivalve tissues (not acidified) were analyzed for carbon and nitrogen elemental and isotopic composition (C, N, $\delta^{13}C$, $\delta^{15}N$) at the Center for Physical Science and Technology (Vilnius, Lithuania) with a Flash EA 1112 Series Elemental Analyzer (Thermo Finnigan) connected to a DeltaV Advantage Isotope Ratio Mass Spectrometer (Thermo Fisher). Carbon and nitrogen elemental content are expressed in percent (%) of mass ratio (g g$^{-1}$) dry weight, while C:N ratio is express in mol mol$^{-1}$. Isotope ratios are expressed using the conventional delta notation: $\delta^{13}C_{sample}$ or $\delta^{15}N_{sample} = [(R_{sample}/R_{standard}) - 1]$, where R = $^{13}C/^{12}C$ or $^{15}N/^{14}N$, in per mil deviation (‰) from international reference, Vienna Pee Dee Belemnite for $\delta^{13}C$ and atmospheric $N^2$ for $\delta^{15}N$. Mass spectrometer was calibrated against external certified standards (Caffeine IAEA600: $\delta^{13}C$ -27.77 ‰; $\delta^{15}N$ 1.00 ‰, Graphite USGS24: $\delta^{13}C$ -16.05 ‰), and internal standards (casein: $\delta^{13}C$ -23.30 ‰; $\delta^{15}N$ 6.30 ‰, glycine: $\delta^{13}C$ -45.20 ‰; $\delta^{15}N$ 3.00 ‰) controlled using sucrose IAEA-CH-6, ammonium sulfate IAEA-N-1 and IAEA-N-2, Graphite USGS24 were added before and every 10 samples within each batch of samples to control analytical performance.

Analytical precision was always better than 0.2 ‰ for $\delta^{13}C$ and $\delta^{15}N$ and 0.1 mol mol$^{-1}$ for C:N ratio (median of the standard deviations of each standard used for all batches: 0.12 ‰ for $\delta^{13}C$ and $\delta^{15}N$ and 0.04 mol mol$^{-1}$ for C:N ratio). Each sample was usually analyzed once and for some peculiar values that seemed too high/low or outside of the average trend, samples were reanalyzed to verify the values. If values were similar (< 0.5 ‰), we chose the average value of both analyses; otherwise, we

chose the value associated with the best analytical precision. The dataset presented here consists in a single value for a given year at a given station (the standard deviation associated with repeated measures is not presented). One data point was considered as an outlier in regard of its carbon isotope value (reanalysis confirmed the value): the station 14-PALL in 2014 (Figure 2) exhibited unexpected low values of $\delta^{13}C$ (-26.96 ‰) and high $\delta^{15}N$ (10.26 ‰) compare to the average values of this station ($\delta^{13}C$ = -18.71 ± 0.75 ‰; $\delta^{15}N$ = 8.41 ± 0.93 ‰), suggesting potential issues in the sampling and/or storing. This value

was therefore removed from average calculations and statistical tests. Within the C and N % datasets, there was a few exceptionally high/low values (Figure 2) for the stations 1-AMBL in 1990 (low C % only), 5-CLHV in 1994 (high C and N %), 18-BOUR in 1990 (low C and N %), 20-RIVE in 1989 (high C and N %) and 30-STMM in 2016 (high C and N %). These values must be considered with caution when interpreting or using C and N % data and were excluded from statistical calculations (see section 2.4). Nevertheless, these data pairs all led to consistent C:N ratio at these sites and years and were

kept for C:N statistical analyses. All the other values presented in this data set are considered analytically valid and were scientifically validated in regard of expert knowledge of each ecosystem.

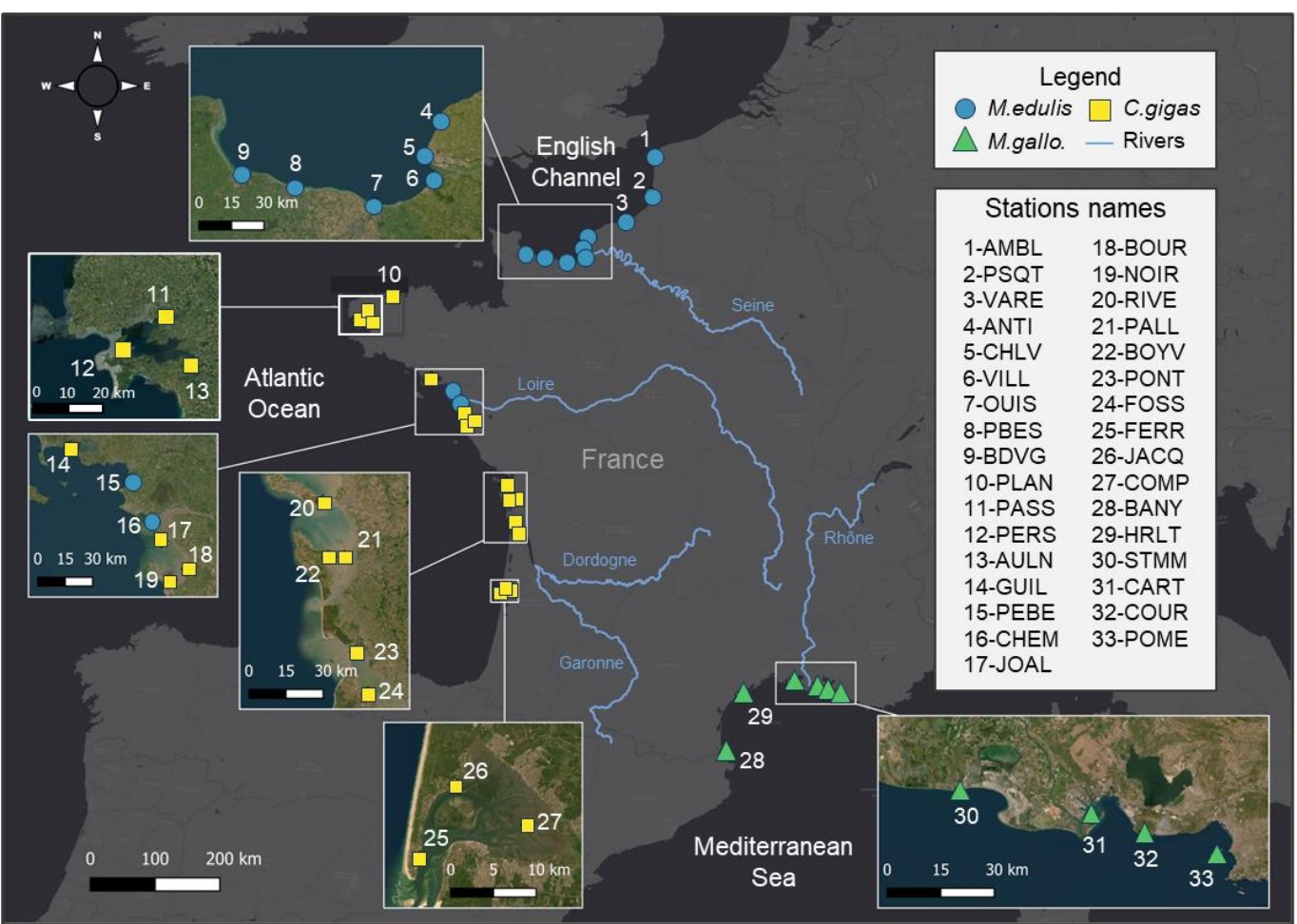

**Figure 1: Map of the 33 stations from the three sea facades of France analyzed for bivalve's elemental and isotopic ratios. Bivalve species are the mussels *Mytilus edulis* and *Mytilus galloprovincialis* and the oyster *Crassostrea gigas*. The main rivers are presented in blue. Basemap source: Esri, HERE, © OpenStreetMap contributors, and the GIS User Community.**

**Table 1: Metadata description for the 33 stations analyzed for bivalve's carbon and nitrogen elemental and isotopic ratios. Species studied are the mussels *Mytilus edulis* and *Mytilus galloprovincialis* and the oyster *Crassostrea gigas*. Station 31-CART represent two stations merged over time (distant from 720m). The geographic coordinate system is WGS84.**

| Sea Facade | Station ID | Station name | ROCCH codes | latitude | longitude | Species | Time series | n years | Missing years |
|---|---|---|---|---|---|---|---|---|---|
| English Channel | 1-AMBL | Ambleteuse | 002-P-032 | 50.807366667 | 1.595466667 | *M. edulis* | 1982-2021 | 36 | 1984, 1985, 1986, 1990 |
| | 2-PSQT | Pointe de St Quentin | 006-P-009 | 50.2808 | 1.52853333 | *M. edulis* | 1988-2021 | 34 | - |
| | 3-VARE | Varengeville | 008-P-013 | 49.9207688241 | 0.9820028963 | *M. edulis* | 1982-2021 | 37 | 1984, 1985, 1986 |
| | 4-ANTI | Antifer - digue | 010-P-014 | 49.649085114 | 0.1503265738 | *M. edulis* | 1983-2021 | 36 | 1984, 1985, 1986 |
| | 5-CLHV | Cap de la Hève | 010-P-055 | 49.5074135756 | 0.0619953224 | *M. edulis* | 1983-2017 | 29 | 1984, 1986, 1987, 1998, 1999, 2000 |
| | 6-VILL | Villerville | 011-P-005 | 49.4040781596 | 0.1236660472 | *M. edulis* | 1982-2021 | 36 | 1984, 1985, 1986 |
| | 7-OUIS | Ouistreham | 010-P-120 | 49.2940709028 | -0.24800513 | *M. edulis* | 1994-2021 | 27 | 2004 |
| | 8-PBES | Port en Bessin | 013-P-001 | 49.3515665687 | -0.7531834463 | *M. edulis* | 1981-2021 | 37 | 1983, 1984, 1988, 2007 |
| | 9-BDVG | Bdv Grandcamp ouest | 014-P-007 | 49.38633333 | -1.101266667 | *M. edulis* | 1981-2021 | 37 | 1984, 1985, 1992, 1996 |
| | 10-PLAN | Pen al Lann | 034-P-001 | 48.665109 | -3.8944 | *C. gigas* | 1982-2021 | 37 | 1984, 1985, 1986 |
| Atlantic Ocean | 11-PASS | Le Passage (b) | 039-P-007 | 48.391067 | -4.384965 | *C. gigas* | 2001-2021 | 21 | - |
| | 12-PERS | Persuel | 039-P-093 | 48.2938273007 | -4.5500595819 | *C. gigas* | 2001-2021 | 21 | - |
| | 13-AULN | Aulne rive droite | 039-P-124 | 48.281083 | -4.260048 | *C. gigas* | 1982-2021 | 38 | 1984, 1985 |
| | 14-GUIL | Le Guilvin | 060-P-001 | 47.56765 | -2.9338 | *C. gigas* | 1982-2021 | 39 | 1986 |
| | 15-PEBE | Pen Bé | 066-P-003 | 47.4306639649 | -2.4679991263 | *M. edulis* | 1982-2021 | 40 | - |
| | 16-CHEM | Pointe de Chemoulin | 070-P-102 | 47.234632 | -2.297076 | *M. edulis* | 1981-2021 | 39 | 1982, 1983 |
| | 17-JOAL | Joalland (b) | 070-P-006 | 47.1568267957 | -2.2224877474 | *C. gigas* | 2010-2021 | 11 | - |
| | 18-BOUR | Bourgneuf - Coupelasse | 071-P-065 | 47.0123255923 | -2.0229805005 | *C. gigas* | 1984-2021 | 37 | 1985 |
| | 19-NOIR | Noirmoutier - Gresse-loup | 071-P-068 | 46.95066 | -2.146303 | *C. gigas* | 1983-2021 | 38 | 1985 |
| | 20-RIVE | Rivedoux | 076-P-032 | 46.163319 | -1.27077 | *C. gigas* | 1981-2021 | 40 | 1986 |
| | 21-PALL | Les Palles | 080-P-004 | 45.9674774781 | -1.1414394441 | *C. gigas* | 1982-2021 | 39 | 1983 |
| | 22-BOYV | Boyardville | 080-P-033 | 45.9638097847 | -1.2259409199 | *C. gigas* | 1984-2021 | 38 | - |
| | 23-PONT | Pontaillac | 084-P-015 | 45.6251372031 | -1.0560967723 | *C. gigas* | 1984-2021 | 38 | - |
| | 24-FOSS | La Fosse | 085-P-007 | 45.475634584 | -0.9845921143 | *C. gigas* | 1981-2021 | 40 | 2018 |
| | 25-FERR | Cap Ferret | 087-P-013 | 44.6439465896 | -1.2412452987 | *C. gigas* | 1982-2021 | 39 | 2005 |
| | 26-JACQ | Les Jacquets | 088-P-067 | 44.7222821428 | -1.1945794691 | *C. gigas* | 1982-2021 | 38 | 1983, 1985 |
| | 27-COMP | Comprian | 088-P-069 | 44.6839491697 | -1.08457666 | *C. gigas* | 1982-2021 | 40 | - |
| Mediterranean Sea | 28-BANY | Banyuls - Labo Arago | 094-P-008 | 42.4806130241 | 3.1388780428 | *M.galloprovincialis* | 1981-2021 | 32 | 1982, 1986, 1987, 1988, 1989, 1990, 2003 |
| | 29-HRLT | Embouchure de l'Hérault | 095-P-026 | 43.2756325016 | 3.4405360127 | *M.galloprovincialis* | 1985-2021 | 31 | 1986, 1987, 1997, 1998, 2016 |
| | 30-STMM | Les Stes Maries de la mer | 106-P-018 | 43.4439790562 | 4.4205519936 | *M.galloprovincialis* | 1982-2021 | 38 | 1984, 1986 |
| | 31-CART | Anse de Carteau | 109-P-025 / -027 | 43.3756486632 | 4.8755622202 | *M.galloprovincialis* | 1982-2021 | 37 | 1986, 1996, 2007 |
| | 32-COUR | Cap Courronne | 111-P-002 | 43.3239826009 | 5.0539000233 | *M.galloprovincialis* | 1983-2021 | 37 | 1986, 1996 |
| | 33-POME | Pomègues Est | 111-P-025 | 43.2673170789 | 5.3005726112 | *M.galloprovincialis* | 1998-2021 | 24 | - |

**Table 2: Environmental characteristics of the 33 stations studied for bivalves elemental and isotopic ratios. Average values for salinity, water temperature and chlorophyll a were calculated, when available, over the studied period from monitoring stations located nearby bivalves sampling stations, with data were retrieved from Surval (https://surval.ifremer.fr - Ifremer) and Somlit (https://www.somlit.fr/ - INSU) databases. Average annual flow rates of the main rivers were calculated over the study period from the HydroPortail database (https://hydro.eaufrance.fr/). Trophic status was defined for each station based on literature (Liénart et al., 2017; Lheureux et al., 2023) and the knowledge of local experts and reflect the average status over the studied period.**

| Sea Facade | Station ID | Ecosystem | Ecosystem type | Tidal range | Salinity (mean±sd) | Water temperature (°C) (mean±sd) | Chlorophyll $a$ (µg L-1) (mean±sd) | Trophic status | Main river influence | Annual flow rates (m$^3$ s$^{-1}$) (mean±sd) | Additional local river influence |
|---|---|---|---|---|---|---|---|---|---|---|---|
| English Channel | 1-AMBL | Eastern English Channel | Littoral | Megatidal | 34 ± 1,0 | 12,7 ± 4,4 | 4,8 ± 4,6 | Eutroph | Seine | 499 ± 332 | Liane |
| | 2-PSQT | Eastern English Channel | Littoral | Megatidal | 33 ± 1,5 | 13,0 ± 4,7 | 8,2 ± 7,8 | Eutroph | Seine | 449 ± 332 | Somme |
| | 3-VARE | Eastern English Channel | Littoral | Megatidal | 33 ± 1,1 | 13,4 ± 4,5 | 2,2 ± 2,3 | Eutroph | Seine | 449 ± 332 | - |
| | 4-ANTI | Bay of Seine | Open bay | Macrotidal | 32 ± 1,8 | 14,3 ± 4,3 | 4,8 ± 7,1 | Eutroph | Seine | 449 ± 332 | - |
| | 5-CLHV | Bay of Seine | Estuary mouth | Macrotidal | 29 ± 3,5 | 12,9 ± 4,4 | 4,0 ± 4,7 | Eutroph | Seine | 449 ± 332 | - |
| | 6-VILL | Bay of Seine | Estuary mouth | Macrotidal | 29 ± 3,5 | 12,9 ± 4,4 | 4,0 ± 4,7 | Eutroph | Seine | 449 ± 332 | - |
| | 7-OUIS | Bay of Seine | Open bay | Macrotidal | 33 ± 0,8 | 15,2 ± 4,2 | 5,4 ± 5,2 | Eutroph | Seine | 449 ± 332 | Orne |
| | 8-PBES | Bay of Seine | Open bay | Macrotidal | 34 ± 0,5 | 13,8 ± 4,4 | 2,8 ± 2,9 | Eutroph | Seine | 449 ± 332 | - |
| | 9-BDVG | Bay of Veys | Open bay | Macrotidal | 33 ± 0,7 | 13,8 ± 4,2 | 2,6 ± 2,5 | Eutroph | Seine | 449 ± 332 | Vire |
| | 10-PLAN | Western English Channel | Ria | Macrotidal | 35 ± 0,2 | 12,9 ± 2,2 | 0,9 ± 0,8 | Mesotroph | - | - | - |
| Atlantic Ocean | 11-PASS | Bay of Brest | Semi-enclosed ria | Macrotidal | 27 ± 7,5 | 12,9 ± 4,1 | NA ± NA | Mesotroph | Elorn | 6 ± 5 | - |
| | 12-PERS | Bay of Brest | Semi-enclosed ria | Macrotidal | 35 ± 0,6 | 13,4 ± 2,8 | 1,1 ± 0,9 | Mesotroph | Aulne/Elorn | - | - |
| | 13-AULN | Bay of Brest | Semi-enclosed ria | Macrotidal | 23 ± 8,7 | 12,8 ± 4,4 | 2,3 ± 2,7 | Mesotroph | Aulne | 27 ± 29 | - |
| | 14-GUIL | Morbihan Gulf | Semi-enclosed bay | Macrotidal | 33 ± 2 | 14,4 ± 3,7 | 1,8 ± 2,0 | Mesotroph | Auray | 2.8 ± 3.1 | - |
| | 15-PEBE | Bay of Vilaine | Semi-enclosed bay | Macrotidal | 32 ± 3 | 14,8 ± 3,9 | 4,6 ± 4,5 | Eutroph | Vilaine | 27 ± 34 | - |
| | 16-CHEM | Loire Estuary | Estuary mouth | Macrotidal | 31 ± 3,6 | 13,9 ± 3,6 | 3,4 ± 4,9 | Mesotroph | Loire | 833 ± 688 | - |
| | 17-JOAL | Loire Estuary | Estuary mouth | Macrotidal | 31 ± 3,6 | 13,9 ± 3,6 | 3,4 ± 4,9 | Mesotroph | Loire | 833 ± 688 | - |
| | 18-BOUR | Bay of Bourgneuf | Semi-enclosed bay | Macrotidal | 33 ± 2,3 | 13,8 ± 3,9 | 2,7 ± 3,9 | Mesotroph | Loire | 833 ± 688 | - |
| | 19-NOIR | Bay of Bourgneuf | Semi-enclosed bay | Macrotidal | 33 ± 2,3 | 13,8 ± 3,9 | 2,7 ± 3,9 | Mesotroph | Loire | 833 ± 688 | - |
| | 20-RIVE | Charentais Sounds | Semi-enclosed sound | Macrotidal | 33 ± 1,9 | 15,1 ± 4,3 | 2,2 ± 2,2 | Mesotroph | Sèvre Niortaise | 12 ± 13 | - |
| | 21-PALL | Charentais Sounds | Semi-enclosed sound | Macrotidal | 27 ± 7,1 | 14,5 ± 4,8 | 4,6 ± 7,0 | Mesotroph | Charente | 65 ± 66 | Sèvre Niortaise |
| | 22-BOYV | Charentais Sounds | Semi-enclosed sound | Macrotidal | 33 ± 2,5 | 14,9 ± 4,3 | 3,2 ± 4,1 | Mesotroph | Charente | 65 ± 66 | Sèvre Niortaise |
| | 23-PONT | Gironde Estuary | Estuary mouth | Macrotidal | 26 ± 5 | 15,9 ± 4,2 | 1,9 ± 1,3 | Eutroph | Garonne / Dordogne | 539 ± 404 | - |
| | 24-FOSS | Gironde Estuary | Estuary mouth | Macrotidal | 26 ± 5 | 15,9 ± 4,2 | 1,9 ± 1,3 | Eutroph | Garonne / Dordogne | 259 ± 199 | - |
| | 25-FERR | Arcachon Lagoon | Semi-enclosed lagoon | Mesotidal | 34 ± 0,8 | 15,2 ± 3,7 | 1,7 ± 1,1 | Mesotroph | - | 17 ± 14 | - |
| | 26-JACQ | Arcachon Lagoon | Semi-enclosed lagoon | Mesotidal | 31 ± 2,8 | 15,8 ± 5,3 | 2,1 ± 1,1 | Mesotroph | Leyre | 17 ± 14 | - |
| | 27-COMP | Arcachon Lagoon | Semi-enclosed lagoon | Mesotidal | 31 ± 2,8 | 16,0 ± 5,3 | 1,8 ± 1,0 | Mesotroph | Leyre | 17 ± 14 | - |
| Mediterranean Sea | 28-BANY | Gulf of Lion | Open bay | Microtidal | 38 ± 0,7 | 16,7 ± 3,9 | 0,6 ± 0,6 | Oligotroph | - | - | Tech / TêT |
| | 29-HRLT | Gulf of Lion | Open bay | Microtidal | 37 ± 2,4 | 17,7 ± 4,6 | NA ± NA | Oligotroph | Hérault | 35 ± 42 | - |
| | 30-STMM | Gulf of Lion | Open bay | Microtidal | NA ± NA | NA ± NA | NA ± NA | Oligotroph | Rhône | 1669 ± 782 | - |
| | 31-CART | Gulf of Lion | Semi-enclosed bay | Microtidal | 33 ± 3,7 | 16,5 ± 4,5 | 1,0 ± 1,1 | Oligotroph | Rhône | 1669 ± 782 | - |
| | 32-COUR | Gulf of Lion | Open bay | Microtidal | 38 ± 0,3 | 17,0 ± 3,5 | 0,4 ± 0,4 | Oligotroph | Rhône | 1669 ± 782 | - |
| | 33-POME | Gulf of Lion | Open bay | Microtidal | 38 ± 0,3 | 17,0 ± 3,5 | 0,4 ± 0,4 | Oligotroph | Rhône | 1669 ± 782 | - |

## 2.4 Statistical analyses

Statistical analyses were performed with the R software (R Core Team, 2022, 4.3.1 version).

Due to the nature of the dataset (i.e., only one of the three species was sampled for each sea façade, except for stations 15-PEBE and 16-CHEM from the Atlantic facade), it was not possible to test whether the observed differences in elemental and isotopic ratios were due to difference between species or sea facade, nor was it possible to ascertain if there was any interaction

between these two factors. Consequently, we have tested the effects of both factors (species and sea facade) independently. Therefore, particular attention should be given in interpreting the results of these tests. The effects of species (3 levels) and sea façade (3 levels) were tested independently for each elemental and isotopic variables over the entire dataset (all stations and time series) with non-parametric Kruskal-Wallis tests followed by Dunn *post-hoc* tests (R-package 'PMCMRplus' version 1.9.10, function 'kruskalTest()' and 'kwAllPairsDunnTest(), Pohlert, 2023) since normality and homogeneity of variance (inspected using Shapiro-Wilk and Levene tests respectively) were not met, precluding the use of ANOVAs.

Monotonous temporal trends in bivalve elemental and isotopic times series were examined for each station using Mann-Kendall tests corrected for autocorrelation (R-package 'modifiedmk' version 1.6, function 'mmkh()', Patakamuri and O'Brien, 2021). Linear models (R-package 'stats' version 4.2.1, function 'lm()', R Core Team, 2022) were applied to calculate the value of the slope (in ‰ decade$^{-1}$) for each time series (note that all the time series do not have the same number of years). Complete time series were tested for shifts allowing for the detection of one unique shift per time series (R-package 'cpm' version 2.3, function 'detectChangePoint()', Gordon, 2015).

## 3 Data characteristics and general overview

This dataset illustrates the temporal and spatial variability of carbon and nitrogen content (C and N %) and elemental and isotopic ratios (C:N, $\delta^{13}$C, $\delta^{15}$N) of three species of bivalve along the French coasts from 1981 to 2021. Overall, the average $\delta^{13}$C and $\delta^{15}$N values of bivalves were significantly lower for the stations of the Mediterranean Sea compared with the stations of the English Channel and the Atlantic Ocean. We observed that stations closest to river mouths displayed the lowest average $\delta^{13}$C values compared to those more distant from river influence. Over the last 10 to 40 years, nearly all stations exhibited a significant decrease in $\delta^{13}$C and $\delta^{15}$N. There was no spatial nor temporal pattern in C:N but values differed between bivalve genus (mussel vs oyster). It is important to note that, due to the nature of the dataset (one species present for each sea façade), it was not possible across all our study sites, to determine whether the differences observed in elemental and isotopic ratios between the three bivalve species and across the three sea facades were due to species effects or spatial differences. However, this could be assessed at the scale of the Atlantic facade (see sections 2.4 and 3.1).

### 3.1 Taxonomic patterns

Different species may exhibit different C and N elemental and isotopic ratios under the same environmental and growth conditions (Mele et al., 2023), and within a single species, varying environmental conditions can lead to difference in these values (e.g. Magni et al., 2012; Briant et al., 2018). Overall, bivalve $\delta^{13}$C ranged from -23.29 to -16.98 ‰, $\delta^{15}$N from 2.47 to 13.15 ‰, C:N from 4.13 to 8.12 mol mol$^{-1}$, and C from 29.81 to 48.64 % and N from 5.81 to 10.57 % (out of extreme data, see 2.3). There was a significant difference in $\delta^{13}$C and C content (p < 0.001) between *M. galloprovincialis* ($\delta^{13}$C: -20.05 ± 0.98 ‰; C: 36.10 ± 1.53 %) and the other two species *M. edulis* ($\delta^{13}$C: -19.38 ± 1.04 ‰; C: 39.80 ± 1.93 %) and *C. gigas*

(δ¹³C: -19.25 ± 1.06 ‰; C: 39.30 ± 1.90 %), which were not significantly different (Figure 2). There was a significant difference in N content (p < 0.001) between *C. gigas* (8.02 ± 0.81 %) and the two other species *M. edulis* (9.16 ± 0.46 %) and *M. galloprovincialis* (9.06 ± 0.44 %; Figure 2). Finally, there was a significant difference in δ¹⁵N (p < 0.001) between the three species of bivalves (*C. gigas*: 8.42 ± 1.09 ‰; *M. edulis* 9.41 ± 1.19 ‰; *M. galloprovincialis* 5.70 ± 1.49 ‰) as well as in C:N (p < 0.001; *C. gigas*: 5.77 ± 0.62 mol mol⁻¹; *M. edulis* 5.07 ± 0.31 mol mol⁻¹; *M. galloprovincialis* 4.65 ± 0.23 mol mol⁻¹;

Figure 2). Since the mussel *M. galloprovincialis* is exclusively present in the Mediterranean Sea and the two other species, *M. edulis* and *C. gigas* are mostly present in the English Channel and Atlantic Ocean, respectively, it is difficult to separate the species factor from the spatial factor (see 3.2) in elemental and isotopic ratios (especially in δ¹³C where *M. galloprovincialis* differs significantly). However, within the Atlantic facade where mostly oysters are sampled (15/17 stations), two stations are sampled for mussels (stations 15-PEBE and 16-CHEM), hence species effect can be tested for this facade. There was no

significant differences in δ¹³C between the two species (*C. gigas*: -19.24 ± 1.07 ‰; *M. edulis*: -19.36 ± 1.02 ‰), but δ¹⁵N, C and N contents, and C:N ratio differed significantly (p < 0.001) between *C. gigas* (δ¹⁵N: 8.43 ± 1.12 ‰; C: 39.40 ± 1.93 %; N: 8.06 ± 0.82 %; C:N: 5.76 ± 0.62 mol mol⁻¹) and *M. edulis* (δ¹⁵N: 10.54 ± 0.88 ‰; C: 38.20 ± 1.33 %; N: 9.34 ± 0.39 %; C:N: 4.78 ± 0.19 mol mol⁻¹). Note that the unequal sample sizes (15 stations for oyster, 2 stations for mussels) can reduce the power of the test and the ability to detect real differences.

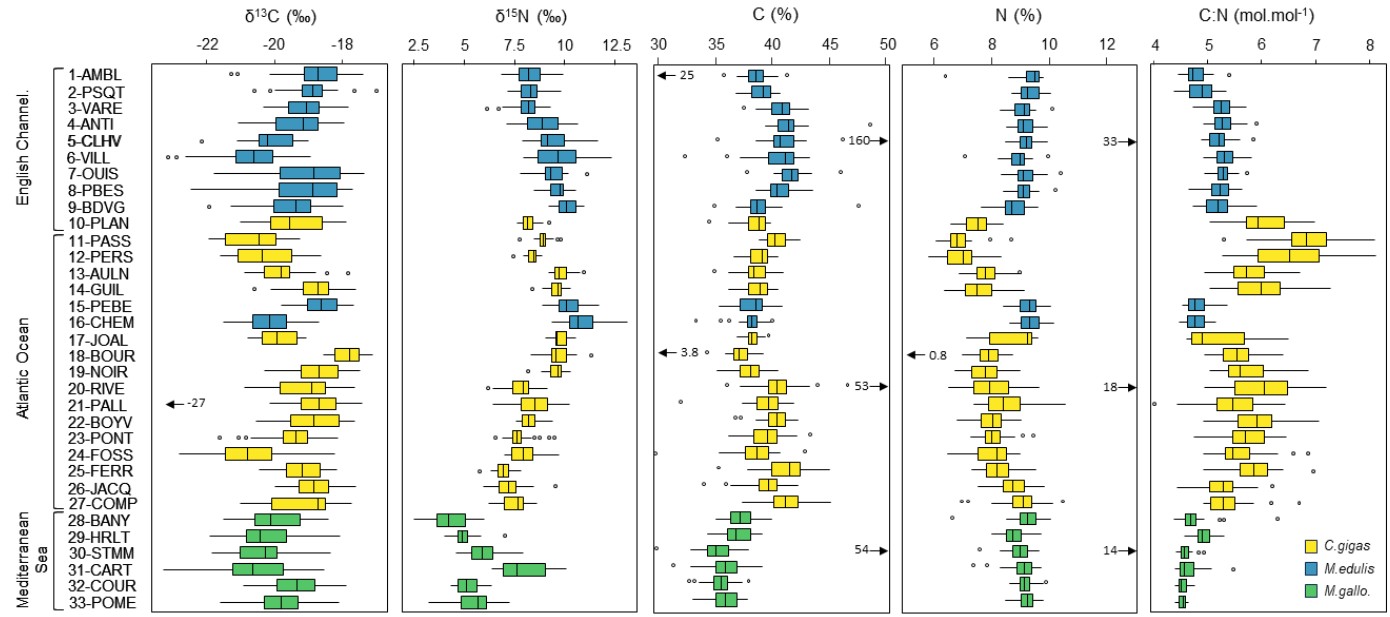

**Figure 2: Boxplots of δ¹³C, δ¹⁵N, C, N and C:N values of bivalves for the 33 stations of the three French sea facades over the period 1981-2021 (*Crassostrea gigas*, *Mytilus edulis*, *Mytilus galloprovincialis*). The lower and upper edges of the box indicate the 25th and 75th percentiles and the middle line the median. The whiskers indicate the maximum/minimum values and the open circle the extreme values. The black arrows and associated values represent extreme values out of graphical bounds.**

## 3.2 Spatial patterns

### 3.2.1 Among sea facade

The three sea facades showed significant difference in bivalves $\delta^{13}C$ (p < 0.001). The Mediterranean Sea had lower values (-20.05 ± 0.98 ‰), than the other two sea facades, the English Channel (-19.39 ± 1.04 ‰) and the Atlantic Ocean (-19.26 ± 1.06 ‰) which were not significantly different (Figure 2). There was also a significant difference in bivalve $\delta^{15}N$ among the three sea facades (p < 0.01) showing a latitudinal gradient with high $\delta^{15}N$ values in the English Channel (9.02 ± 1.07 ‰), intermediate values in the Atlantic Ocean (8.71 ± 1.30 ‰), and low values in the Mediterranean Sea (5.70 ± 1.49 ‰; Figure 2). Bivalve C content and C:N ratio were also significantly different among the three sea facades (p < 0.001), with high values in the English Channel (C: 40.0 ± 1.89 %; CN: 5.24 ± 0.42 mol mol$^{-1}$) and Atlantic Ocean (C: 39.20 ± 1.90 %; C:N: 5.63 ± 0.67 mol mol$^{-1}$) and low values in the Mediterranean Sea (C: 36.10 ± 1.53 %; C:N: 4.65 ± 0.23 mol mol$^{-1}$; Figure 2). Finally, N content differed significantly (p < 0.001) between the Atlantic Ocean (8.23 ± 0.89 %) and the two other sea facades (English Channel: 8.94 ± 0.69 %; Mediterranean Sea: 9.06 ± 0.44 %), which is likely due to species effect (oysters vs mussels, see 3.1).

### 3.2.2 Among stations

The lowest $\delta^{13}C$ values were mostly observed in bivalves from the stations inside or close to the main river mouths: 5-CLHV (-20.13 ± 0.70 ‰) and 6-VILL (-20.60 ± 1.03 ‰) for the Seine river; 16-CHEM (-20.15 ± 0.74 ‰) and 17-JOAL (-19.90 ± 0.60 ‰) for the Loire river; 24-FOSS (-20.71 ± 1.04 ‰) for the Gironde estuary; 30-STMM (-20.34 ± 0.81 ‰) and 31-CART (-20.58 ± 1.14 ‰) for the Rhône river (Figure 2). The lowest $\delta^{15}N$ values were mostly observed for bivalve from the Mediterranean Sea stations (all below < 6 ‰) with minimum average values for station 28-BANY (4.27 ± 0.94 ‰) and maximum for station 30-STMM (5.91 ± 0.75 ‰). The highest $\delta^{15}N$ values were observed for stations under the Loire and Seine River plumes (> 9 ‰) e.g., 16-CHEM (10.90 ± 0.90 ‰), 17-JOAL (9.79 ± 0.43 ‰), 5-CLHV (9.41 ± 1.08 ‰), 6-VILL (9.78 ± 1.26 ‰), and locally under the influence of smaller rivers (Figure 2), e.g., 9-BDVG (10.16 ± 0.52 ‰), 13-AULN (9.81 ± 0.42 ‰), 15-PEBE (10.19 ± 0.71 ‰). This pattern is less clear for the Garonne and Dordogne rivers (Gironde estuary) where $\delta^{15}N$ shows intermediate values (23-PONT (7.74 ± 0.60 ‰), 24-FOSS (8.04 ± 0.73 ‰)). Bivalve C and N content and C:N ratio did not show any clear spatial pattern between stations and seemed mostly linked with genus/species effects (differing between facades, Figure 2; see also 3.1).

Interestingly, for stations 16-CHEM and 17-JOAL, both located at the mouth of the Loire river and sampled for mussels and oysters, respectively, the $\delta^{13}C$ signal did not differ significantly (mussels: -19.95 ± 0.74 ‰; oysters: -19.90 ± 0.60 ‰) over the same time period (2010-2021), despite being different species and genus. As for the comparison among the Atlantic façade (see 3.1), $\delta^{15}N$ values significantly differed between the two species (mussels: 10.30 ± 0.44 ‰; oysters: 9.79 ± 0.43 ‰), however, the C:N ratio was not significantly different between the two species for this specific area (mussels: 4.74 ± 0.19 ‰;

oysters: 5.16 ± 0.64 ‰). This suggests that spatial location could be the main driver of bivalve carbon isotope signal but that nitrogen is likely influenced by both global and local processes, including bivalve physiology, that compensate for species difference.

### 3.2.3 Key messages

Bivalves' elemental and isotopic ratios were generally lower for the stations of the Mediterranean Sea compared to those

located in the Atlantic Ocean and English Channel. We found significant differences in carbon and nitrogen content and elemental and isotopic ratios between species (see 3.1), each of which is present in only one sea facade (with a few exceptions), nevertheless, this difference is also linked with spatial variability. Indeed, C and N baselines (for isotope signal) and trophic status (i.e., nutrient and chlorophyll amounts) vary between each sea facade (oligotrophic Mediterranean Sea vs. meso- to eutrophic Atlantic Ocean and English Channel; Table 2) but also locally (i.e., proximity to river mouth). To overcome the lack

of nutrients available in the Mediterranean Sea, some specific groups of phytoplankton (i.e., diazotrophs) uses $^{15}$N-depleted atmospheric nitrogen as N source which decreases both the $\delta^{15}$N of the overall particulate organic matter (Kerhervé et al., 2001; Wannicke et al., 2010; Landrum et al., 2011; Liénart et al., 2017) and, by trophic propagation, the low $\delta^{15}$N signal is reflected in bivalves (Liénart et al., 2022, 2023). There is a clear difference in $\delta^{15}$N baseline between the Mediterranean sea and the other French sea facades (Liénart et al., 2017). Generally, there was a clear difference in $\delta^{13}$C signal close to river

mouths, decreasing along river plumes, and in $\delta^{15}$N for some of the main (mostly eutrophicated) rivers. Enriched $^{15}$N signal of nutrients coming from rivers with extensive watershed agricultural activities and urban outlets is reflected in the high $\delta^{15}$N values of bivalves sampled close to river mouths (Fukumori et al., 2008; Thibault et al., 2020). Similarly, low $\delta^{13}$C values for bivalves sampled close to river mouths are mostly linked to the inputs of continental particulate material bearing this specific signal (Liénart et al., 2017). $\delta^{13}$C values of dissolved inorganic carbon (DIC) are lower in riverine/freshwater ecosystems

compared to marine environments (Mook and Rozanski, 2000), resulting in more negative $\delta^{13}$C values in phytoplankton, which may be ingested locally by bivalves. Additionally, $\delta^{13}$C of particulate organic carbon (POC) from terrestrial sources tends to be more negative (Liénart et al., 2017) and can be consumed by bivalves. However, whether terrestrial POC is a significant food source for bivalves remains debated (Malet et al., 2008; Marín Leal et al., 2008; Dubois et al., 2014). Similarly, the $\delta^{15}$N signal of freshwater and marine phytoplankton differs due to the distinct nitrogen sources in these two environments. Therefore,

it is not surprising that the $\delta^{15}$N signal in bivalves varies along a gradient related to proximity to river mouths.

### 3.3 Temporal patterns

#### 3.3.1 Pluri-decadal $\delta^{13}C$ dataset

Bivalves $\delta^{13}C$ showed a significant decrease over the period 1981-2021 for 82 % of the stations (27/33 stations; Figure 3). The Mediterranean Sea exhibited a significant decrease for all stations (6/6 stations), the Atlantic Ocean for 82 % of the stations (14/17 stations) and 70 % of the stations in the English Channel (7/10 stations). The average decrease over the total period was consistent across all facades, averaging -0.58 ± 0.28 ‰ per decade, ranging from -0.22 to -1.21 ‰ per decade (i.e., -4.85 to -0.89 ‰ over the last 40 years) when the slopes were significant. This decrease in bivalves $\delta^{13}C$ was more pronounced for the shorter time series starting from late 1990's (17-JOAL, 2010-2021; 11-PASS and 12-PERS, 2001-2021; 33-POME, 1998-2021; 7-OUIS, 1994-2021; Figure 3). Overall, the slopes were more pronounced for the Mediterranean Sea and for the stations in the western Bay of Seine, i.e., under the influence of the Seine plume (stations 7 to 10). Some of the stations showed strong interannual variability in bivalves $\delta^{13}C$ (e.g., 1-AMBL, 6-VILL, 24-FOSS, Figure 3).

Shifts in bivalve $\delta^{13}C$ time series were detected for 79 % of the stations (26/33 stations), equally represented in each sea facade. The shifts occurred mostly around the year 1999-2000 (6/26 stations) or after, around the years 2006 ± 1 year and 2012 ± 1 year, with no specific pattern per façade. In the Mediterranean Sea most of the shifts occurred before 2000. The earliest shift was in 1989 (31-CART) and two in 1999, the latest in 2014 (15-PEBE). Usually, no shift was detected at the stations close to river mouths, i.e., Seine (5-CLHV, 6-VILL), Loire (16-CHEM, 17-JOAL), Gironde (24-FOSS) estuaries, and at some of the mouths of smaller rivers (e.g., 15-PEBE, Vilaine river; 29-HRLT, Hérault river).

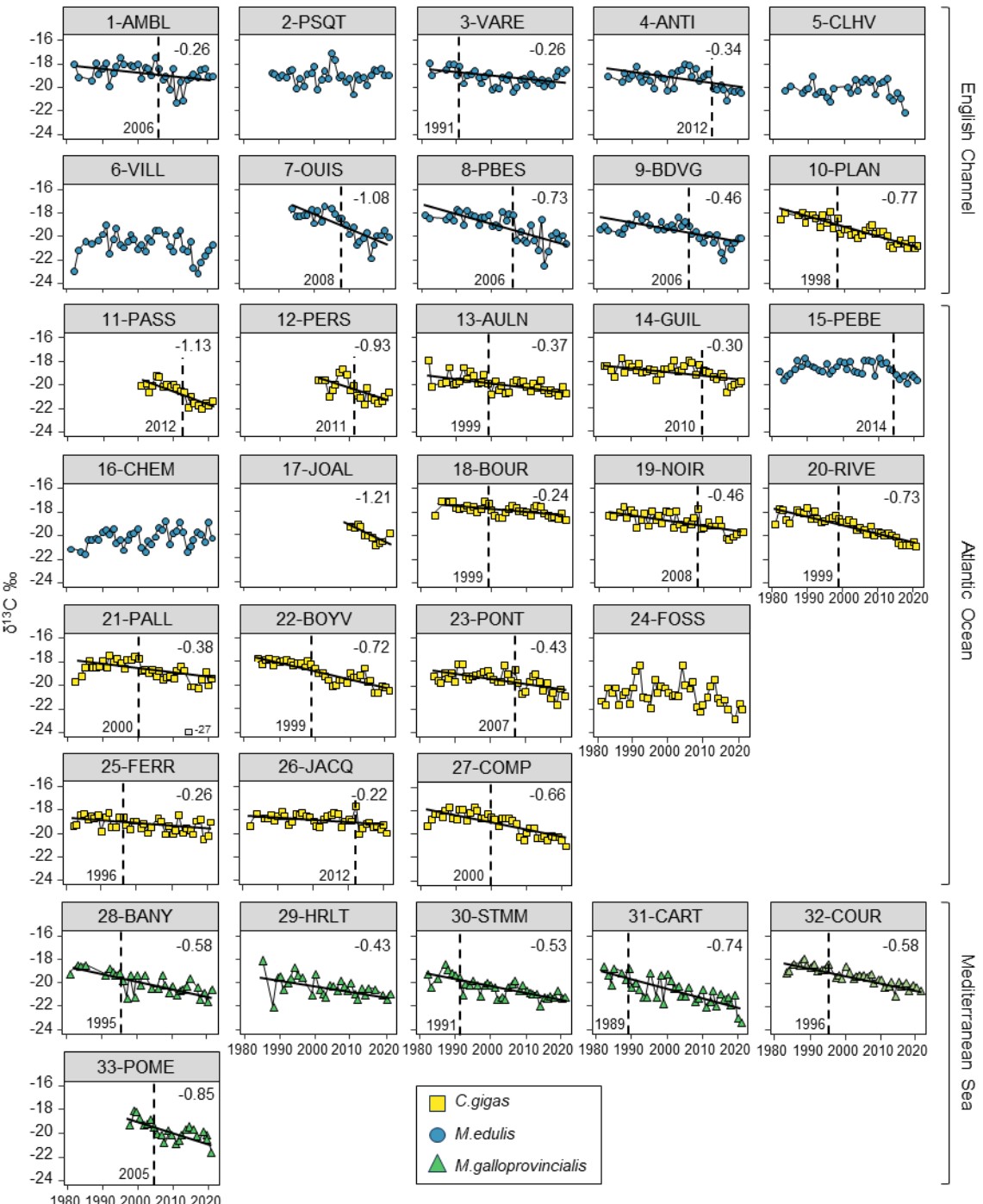

**Figure 3: Bivalves δ¹³C time series over the period 1981-2021 for the 33 stations of the three French sea facades (*Crassostrea gigas*,**
***Mytilus edulis*, *Mytilus galloprovincialis*). Black lines correspond to significant trends (Mann-Kendall tests, p-value < 0.05), the slope value (‰ per decade) appears in the upper right corner. Shifts are represented by the vertical dash line with the year mentioned on its down left. One outlier is shown in grey (21-PALL, -27‰) but was not considered for statistical analyses.**

### 3.3.2 Pluri-decadal $\delta^{15}$N dataset

Bivalve $\delta^{15}$N also showed a significant decrease over the period 1981-2021 for 64 % of the stations (22/33 stations; Figure 4).
The Mediterranean Sea exhibited a significant decrease for 83 % stations (5/6 stations), the English Channel for 80 % of the stations (8/10 stations) and only 53 % of the stations in the Atlantic Ocean (9/17 stations). Only one station showed a significant increase (25-FERR, 0.34 ‰ over 39 years). The average decrease was -0.44 ± 0.20 ‰ per decade, ranging from -0.18 to -0.84 ‰ per decade (i.e., -0.72 to -3.35 ‰ over the last 40 years; Figure 4). However, this decrease was more pronounced in the English Channel (-0.50 ± 0.21 ‰) in the eastern part and close the Seine river mouth (e.g., 3-VARE, 4-ANTI, 6-VILL), and Mediterranean Sea (-0.54 ± 0.20 ‰) compared with the Atlantic Ocean (-0.32 ± 0.08 ‰) where the lowest slopes were near the Loire river mouth and the inner two stations of the Arcachon lagoon. The interannual variability was relatively low for $\delta^{15}$N with the exception of some stations (e.g., 1-AMBL, 6-VILL, 16-CHEM, 33-POME).

Shifts in bivalve $\delta^{15}$N time series were detected for 70 % of the stations (23/33 stations), mostly in the English Channel and Mediterranean Sea. The shifts occurred mostly around the year 1999-2000 (5/23 stations), then around the years 1995-1996 (4/23 stations) and 2004 ± 1 year (4/23 stations), with shifts mostly occurring before 2000 in the Mediterranean Sea. For each sea facade, there were often shifts occurring the same year or in two close years for nearby stations (e.g., 15-PEBE, 2003 and 16-CHEM, 2004). The earliest shift was in 1991 (29-HRLT), the latest in 2013 (14-GUIL).

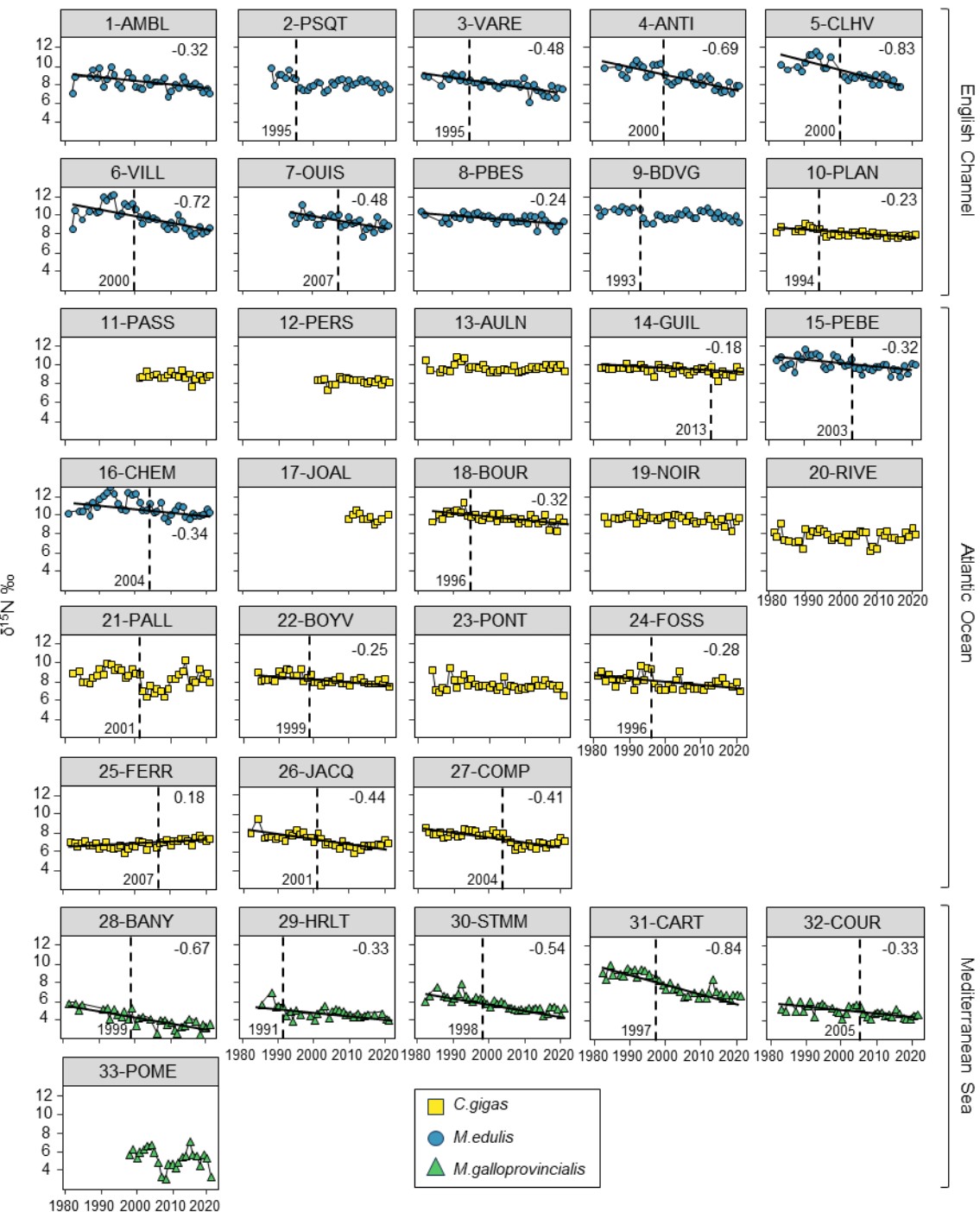

**Figure 4: Bivalves δ¹⁵N time series over the period 1981-2021 for the 33 stations of the three French sea facades (*Crassostrea gigas*, *Mytilus edulis*, *Mytilus galloprovincialis*). Black lines correspond to significant trends (Mann-Kendall tests, p-value < 0.05), the slope value (‰ per decade) appears in the upper right corner. Shifts are represented by the vertical dash line with the year mentioned on its down left.**

### 3.3.3 Pluri-decadal C and N datasets

Bivalve C and N contents and C:N ratio showed significant trends for less than half of the stations (between 30 and 36 %) over the period 1981-2021 (C: 12/33 stations; N: 11/33 stations; C:N: 10/33 stations; Figure 5, 6 and 7). For C content, there was the same number of increasing and decreasing trends (both 18 %, 6/33 stations), with no spatial pattern, while N content showed only decreasing trends (33%, 11/33 stations). However, most of the significant trends were increasing for C:N ratio (24 %, 8/33 stations), exclusively in the Atlantic Ocean (6/33 stations) and the English Channel (2/33 stations) and only 6 % (2/33 stations) of the stations showed decreasing trends, all in the Mediterranean Sea. The average increase in C:N ratio was $0.27 \pm 0.37$ mol mol$^{-1}$ per decade, ranging from 0.05 to 0.38 mol mol$^{-1}$ per decade (i.e., 0.20 to 1.53 mol mol$^{-1}$ over the last 40 years; excluding 17-JOAL with a 1.45 mol mol$^{-1}$ increase for 11 years; Figure 5). The average C:N decrease in the Mediterranean sea was $-0.05 \pm 0.02$ mol mol$^{-1}$ per decade. Interannual variability in bivalve C:N was either very low for the two mussel species (e.g., 7-OUIS, 16-CHEM, 33-POME), or very large for the oysters (e.g., 12-PERS, 19-NOIR, 22-BOYV). The decreasing trends in C:N ratio were mostly linked to decrease in bivalves N % (6/8 stations), whereas only 3/8 stations showed a significant increase in C %. Shifts in bivalve C, N and C:N time series were detected only for 18 to 33 % of the stations (C: 6/33 stations; N: 9/33 stations; C:N: 11/33 stations), without any peculiar spatial pattern. Most of the shifts occurred after the year 2000 (C: 4/33 stations; N: 5/33 stations; C:N: 7/11 stations). The earliest C:N shift was in 1989 (27-COMP), the latest in 2014 (6-VILL, 22-BOYV). There was no apparent spatial pattern in the C, N and C:N shifts.

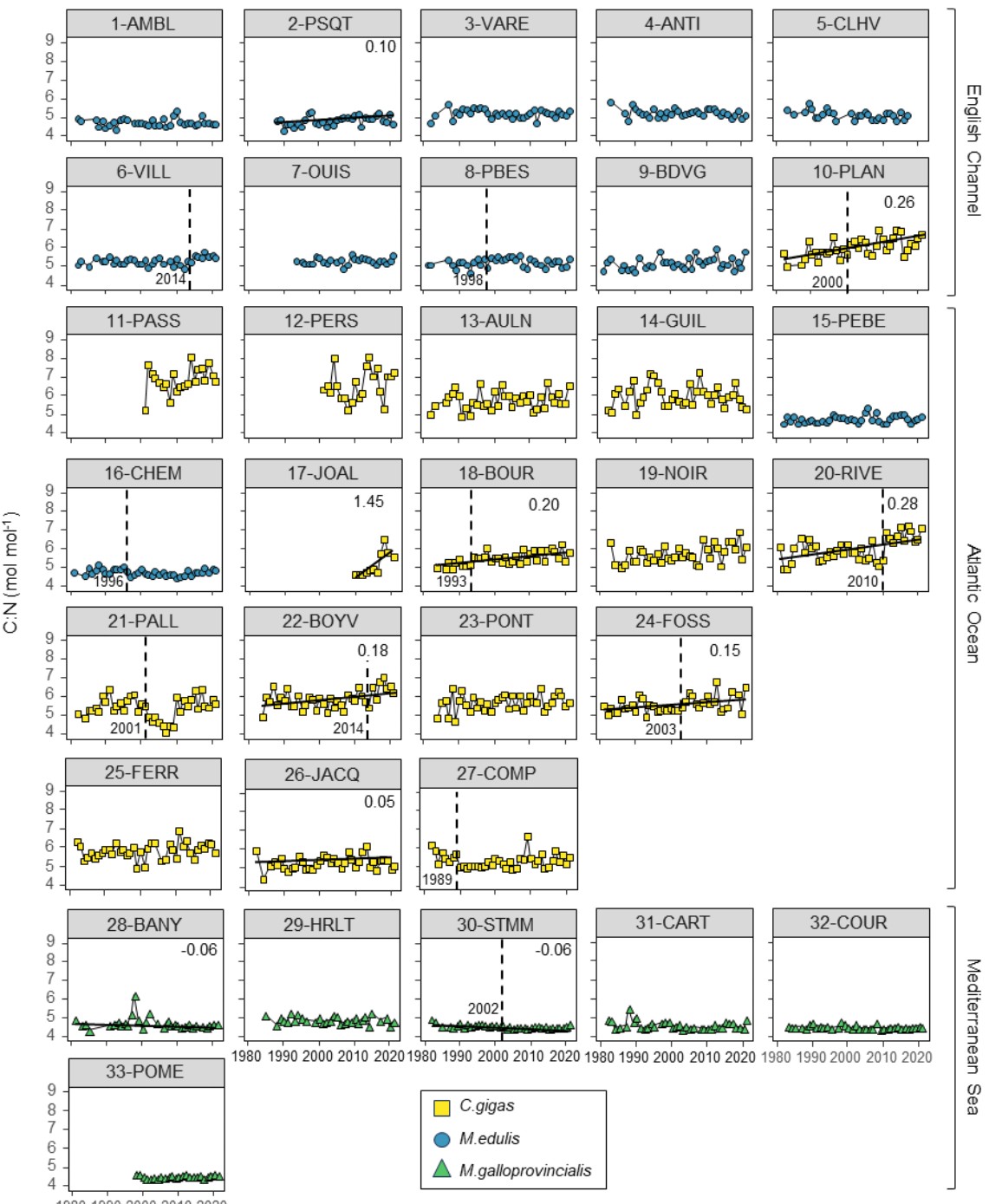

**Figure 5: Bivalves C:N time series over the period 1981-2021 for the 33 stations of the three French sea facades (*Crassostrea gigas*, *Mytilus edulis*, *Mytilus galloprovincialis*). Black lines correspond to significant trends (Mann-Kendall tests, p-value < 0.05), the slope value (‰ per decade) appears in the upper right corner. Shifts are represented by the vertical dash line with the year mentioned on its down left.**

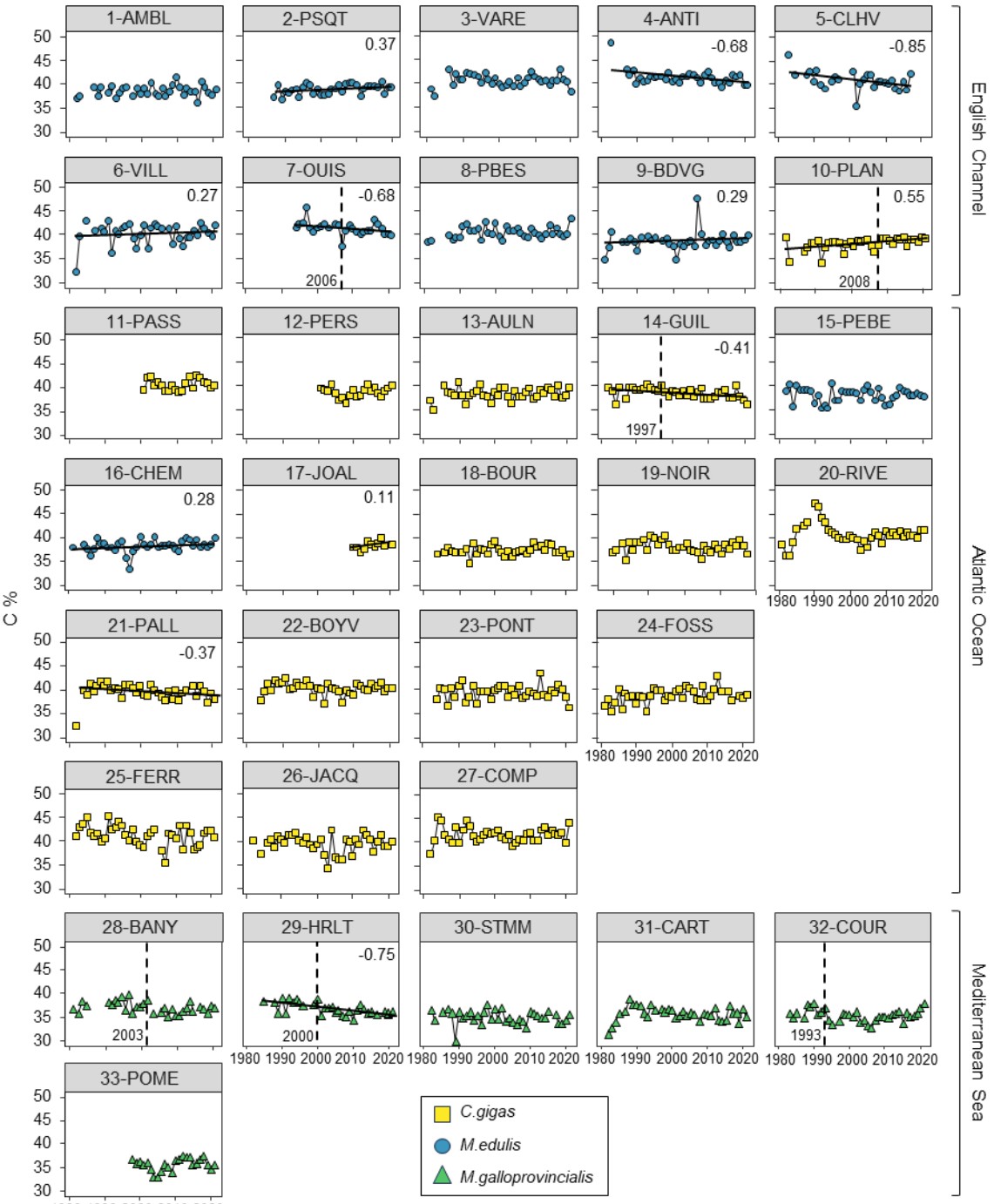

**Figure 6: Bivalves C content (%) time series over the period 1981-2021 for the 33 stations of the three French sea facades (*Crassostrea gigas*, *Mytilus edulis*, *Mytilus galloprovincialis*). Black lines correspond to significant trends (Mann-Kendall tests, p-value < 0.05, calculated excluding extreme values mentioned in section 2.3), the slope value (‰ per decade) appears in the upper right corner.**

**Shifts are represented by the vertical dash line with the year mentioned on its down left. Extreme values mentioned in section 2.3 are not represented and were excluded from the calculations of the slopes and shifts.**

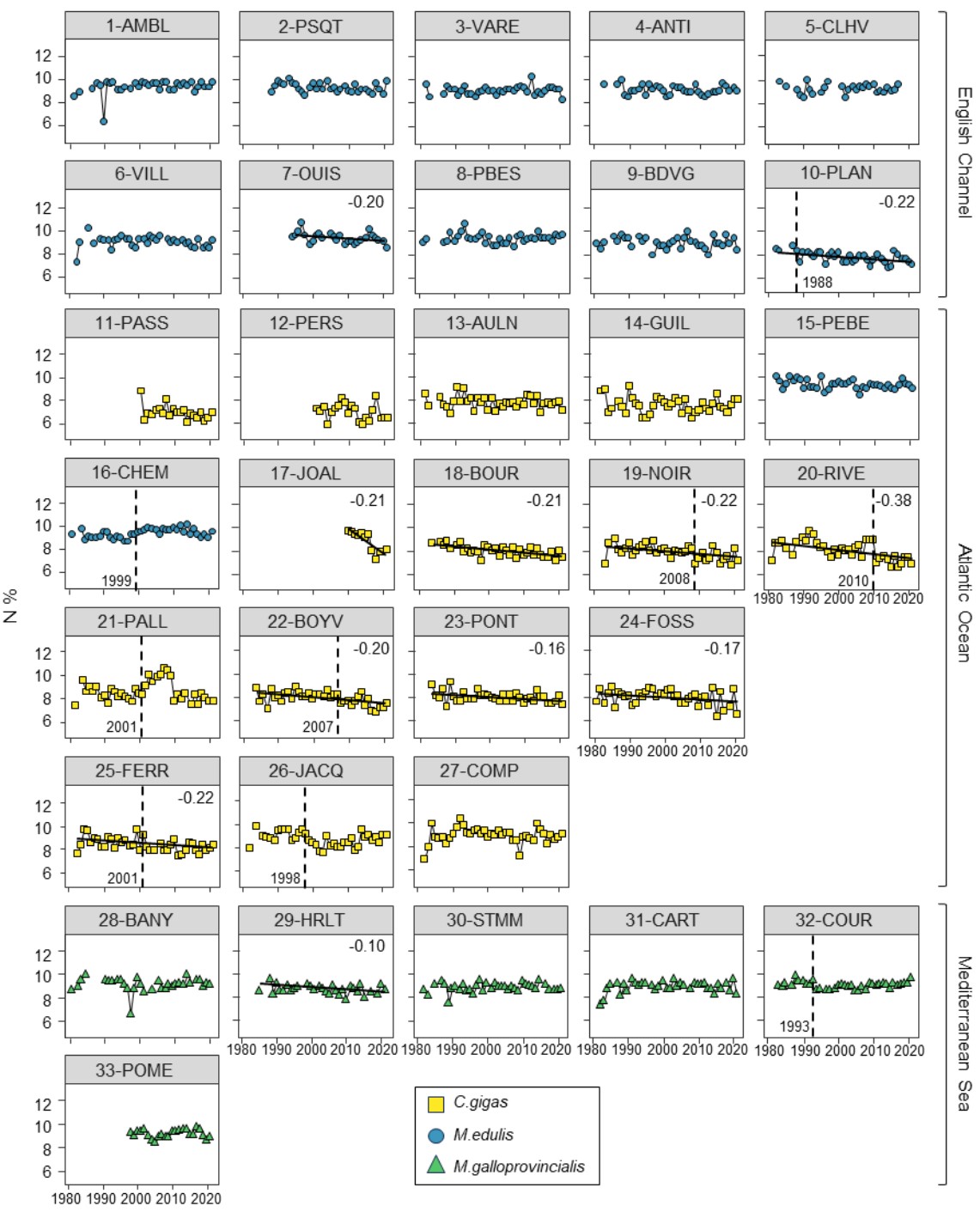

**Figure 7: Bivalves N content (%) time series over the period 1981-2021 for the 33 stations of the three French sea facades (*Crassostrea*** **350** ***gigas*, *Mytilus edulis*, *Mytilus galloprovincialis*). Black lines correspond to significant trends (Mann-Kendall tests, p-value < 0.05,**

**calculated excluding extreme values mentioned in section 2.3), the slope value (‰ per decade) appears in the upper right corner. Shifts are represented by the vertical dash line with the year mentioned on its down left. Extreme values mentioned in section 2.3 are not represented and were excluded from the calculations of the slopes and shifts.**

### 3.3.4 Key messages

In summary, over the last 30 to 40 years, bivalve have shown a general decrease in both $\delta^{13}C$ and $\delta^{15}N$ for most stations of the three French sea facades. These trends are most likely linked with global processes occurring at large spatial scale such as climate change. Liénart et al. (2024) explained the decrease in $\delta^{13}C$ is primarily associated with the increase in atmospheric $CO_2$ concentrations due to fossil-fuel burning, which generates a decrease in atmospheric $CO_2$ $\delta^{13}C$ values occurring since the industrial revolution in 1850 (the Suess effect; Keeling 1979; Gruber et al., 1999). This decrease in $\delta^{13}C$ values propagates

along marine food webs and is visible in organisms' tissues (Schloesser et al. 2009; Liénart et al., 2022, 2024b). However, other global factors such as rising temperatures or climate indices may also contribute to this overall decrease in $\delta^{13}C$ through passive (e.g., Suess effect) or active processes related to bivalve physiology (e.g., change in fractionation; Liénart et al., 2024b). Decreasing $\delta^{15}N$ trends are potentially linked to global factors, indirectly through a general decrease in nutrients and particles inputs from rivers (Milliman et al., 2008; Bauer et al., 2013). Since organic particles generated in areas with high human

activities bears a high $\delta^{15}N$ signal, the decrease of such [15]N-enriched inputs is reflected in coastal water and marine fauna (Connolly et al., 2013). However, such changes are strongly influenced by regional and local factors such as regional climate, watershed activities and damming (Milliman et al., 2008) resulting in different slopes values in isotope time series close to river mouths (more pronounced for both C and N isotopes). The observed decrease in both isotopic ratios likely results from the cumulative, synergistic, or antagonistic effects of global and regional/local influences (Cabral et al., 2019). Such interacting

pressures are more likely to occur in regions strongly affected by climate change, such as the Mediterranean Sea (Tuel and Eltahir, 2020). In this dataset, isotopes reveal both global (climate) effects, mostly through $\delta^{13}C$, and some local variability in the organic matter used by bivalves, as shown by the different $\delta^{13}C$ (low) and $\delta^{15}N$ (high) signals along the river-mouth-to-sea gradient. The decrease in bivalve $\delta^{13}C$ and in $\delta^{15}N$ over the past decades may indicate a decrease in nutrient and particles inputs from the rivers over the same period. Finally, as an indicator of bivalve condition and physiology (Elser et al., 2003), the

increase in C:N ratio observed for most stations in the Atlantic and English Channel can be attributed to either an increase in carbon content (i.e., increase in lipid or carbohydrate content), which is unlikely given rising temperatures and appear significant for only few of our stations, or to a decrease in nitrogen content (i.e., proteins, amino acids), as observed in our data and likely due to a decrease in N-nutrient availability in the environment which is consistent with the observed decrease in $\delta^{15}N$. The absence of trends at most of the stations and the large interannual variability of C:N ratio may result from the

complex interplay between global and local environmental effects and bivalves physiology on this parameter.

**4 Data availability**

The dataset presented in this article and the metadata associated can be freely accessed on the SEANOE open access repository (https://www.seanoe.org/) under the DOI https://doi.org/10.17882/100583 (Liénart et al., 2024a). The data gives the $\delta^{13}$C and $\delta^{15}$N (‰), C and N (%) and C:N ratio (mol mol$^{-1}$), for each station and year, with sampling date and species name. For each station, it contains the sea façade, station ID, station name, ROCCH codes, latitude and longitude (geographic coordinate system WGS84) as presented in Table 1. Carbon isotopic time-series are already published (Liénart et al., 2024b, except from station 17-JOAL) in a study focused on global effect on coastal ecosystems from large-scale anthropogenic and natural climate proxies, including the Suess effect, over the period 1981-2021. Bivalve $\delta^{13}$C data and $\delta^{13}$C corrected from the Suess effect are available under the permanent identifier https://doi.org/10.6084/m9.figshare.24884871.v1 (Liénart et al., 2024c). The companion data from the ROCCH (contaminants) are available on the 'Surval' database (https://surval.ifremer.fr).

**5 Conclusion and recommendation for use**

By providing this dataset to the scientific community including caveats for interpreting such data spatially and over time, we expect it will be useful for numerous ecological studies. Such data is relevant to trace nutrients origin, to set accurate baselines to study organisms diets and food webs structure, and can be and indicator of water quality over space and time through bivalve physiology. It could also provide valuable input for developing predictive models of bivalve physiology (Emmery et al., 2011) or trophic ecology (Marín Leal et al., 2008). Multi-decadal time-series allow scientists to understand coastal ecosystems responses to global change through biological and biogeochemical processes. The carbon isotope dataset is already part of a study where trends are thoughtfully interpreted in regard of global proxies for climate and anthropogenic changes and corrected for the Suess effect (Liénart et al., 2024b). Part of the observed $\delta^{13}$C trends were linked with the Suess effect, leading to shifts in isotope baseline over recent decades (i.e., Suess effect for $\delta^{13}$C) rather than specific changes in ecosystem functioning. Hence, we would like to draw users attention on the need to correct for the Suess effect before comparing biological samples collected one or more decades apart as recommended in literature (Dombrosky 2020; Clark et al. 2021; Liénart et al., 2024b). Additionally, in order to take into account the low $\delta^{13}$C and high $\delta^{15}$N signal observed near river mouths, we advise to compare slope values rather than absolute values when assessing temporal changes between stations. Overall, the complex interplay between global and regional/local effects needs to be considered when interpreting time series. This long-term dataset of elemental and isotopic values in suspension-feeders tissues provides insights into ecosystem dynamics and hold broader significance for advancing scientific understanding in the face of ongoing environmental challenges.

**Author contribution**

CL and NS are leaders of the project. NS, CL, AL, AGP, PLM, XdM, HB, SD and AlG participated in defining the scientific strategy and selection of archived sampled for analysis. AF and NB prepared the samples for analysis. AnG carried out sample

analysis. PLM and AGP are the curator of the ROCCH sample archive. CL and AF performed the statistical analysis. CL prepared the manuscript and figures and integrated the final contribution from all co-authors.

**Competing interests**

The authors declare that they have no competing interests.

**Disclaimer**

Publisher's note: Copernicus Publications remains neutral with regard to jurisdictional claims in published maps and institutional affiliations.

**Acknowledgements**

This article is based on a collaborative work with the team of the ROCCH monitoring network coordinated by Ifremer. We thank the members of the ROCCH and of the different institutions, from the field workers and sample analysts to the coordinators, who made it possible to use these samples.

**Financial support**

This research was funded by the Office Français de la Biodiversité (OFB) within the frame of the research project EVOLECO-BEST.

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
