# Peer review of "Bivalve monitoring over French coasts: multi-decadal records of carbon and nitrogen elemental and isotopic ratios as ecological indicators of global change"

_Earth System Science Data, 2024_

## Author Response (AR1)

**Response to R1 comments**

The article 'Bivalve monitoring over French coasts: multi-decadal records of carbon and nitrogen elemental and isotopic ratios ($\delta^{13}$C, $\delta^{15}$N and C:N) as ecological indicators of global change' presents an extensive long-term study of the carbon and nitrogen isotopes in mussels and oysters over 33 sites along the French coast from 1981 to 2021. The study has used this extensive historic sample set to monitor changing coastal environments, and to establish potential implications for trophic transfer of nutrients into mollusc tissues. This study has been able to use mussel and oyster samples collected by Réseau d'Observation de la Contamination CHimique as bioindicators for chemical contamination to study the nutrient changes over decades using mussel and oyster tissue isotopes. This is an innovative use of existing historic samples to understand environmental change and implication for ecosystem cycling. This is a really nice study and important for considering future environmental change with climate change induced extreme events, increased rainfall, warming all influences the C/N at the coast and uptake for the molluscs living in those environments. The importance of the study refers to using the C/N isotopes to monitor food webs for effective management in a changing world. I would have liked to have seen more discussion on the influence of freshwater inputs at the coastal sites to alter C/N and implications for algal blooms under climate change. Please see below specific points where this could be expanded.

We thank the reviewer for their valuable comments and suggestions which have helped improve the quality of our article. As this article is a data paper, in-depth discussion of the processes associated to the data was not our primary objective. However, we believe it is important to provide key messages and possible interpretations, which we have included accordingly. Therefore, we have expanded the manuscript with a few sentences, particularly addressing the influence of freshwater inputs, as suggested. These data, along with additional environmental data, will be thoroughly discussed in a forthcoming publication that will include environmental databases and examine the effects of global climate and regional factors like river inputs and food availability), as well as the potential effects of bivalve species and physiology on the isotope signal.

Some questions to the authors:

Results

**Line 174** 'it is impossible to determine if the differences observed in elemental and isotopic values between the three species of bivalves and between the three sea facades are due to species effect or spatial differences (see 2.4 and details hereafter).' Can the authors confirm how many of each species were analysed for C/N? As this can vary between individuals. Could this be a factor in the variability? It is known from other studies examinng environmental change e.g. Mele et al., (2023), Lee et al., (2021), Fitzer et al., (2019), Lu et al., (2018) that C/N does differ between species of mussel and oyster and again will differ due to environmental change, i.e. freshwater input at coastal regions and change to algae species can reduce by upto –5 per mil.

Could the authors please discuss this difference in species and environment and can refer to examples to support this. E.g. in reference to previously published data to support this, e.g. ~-20 per mil C for mussels *Mytilus edulis* (Mele et al., 2023, Lee et al., 2021; Lu et al., 2018), oysters <-21 *C. gigas* (Mele et al., 2023), *Saccostrea glomerata* ~-21 - -26 (Fitzer et al., 2019). Nitrogen also differs lighter ~<7 per mil for mussels Mytilus edulis and > ~8 per mil for oysters C. gigas (Mele at al., 2023).

This dataset presents the C and N elemental and isotope values for three bivalve species: the oyster *Crassostrea gigas*, the mussel *Mytilus edulis*, and the mussel *Mytilus galloprovincialis*. The sampling strategy is described in section 2.1, with details regarding the size and number of individuals sampled for each species are provided in lines 84 to 87 : "A minimum of 50 mussels or 10 oysters was required to constitute a representative pooled sample accounting for inter-individual variability". Since the sampling protocol was standardized across sites and years, and each sample is a composite of multiple individuals, inter-individual variability is integrated withing each sample. Therefore, we considered this approach sufficient to minimize intra-species variability in elemental and isotope signals.

Additionally, we acknowledge the intrinsic differences in isotope values between mussels and oysters. The differences observed between species in N isotopes and in the C:N elemental ratio may be attributed to species effects (discussed in section 3.1), which were tested for the Atlantic coast as explained in lines 185-189. We recommend that data users be cautious when making comparisons between species. However, for each species, the selected stations represent a range of environmental conditions (e.g., near river mouths, littoral zones, bays), allowing for comparisons of elemental and isotope values and enabling the analysis of environmental changes on bivalve isotope values.

The sentence on lines 174-175 will be modified as following: "It is important to note that, due to the nature of the dataset (with one species present for each sea façade), it was not possible across all our study sites, to determine whether the differences observed in elemental and isotopic values between the three bivalve species and across the three sea façades were due to species effects or spatial differences. However, this could be assessed at the scale of the Atlantic façade (see sections 2.4 and 3.1).

An introductory sentence will be added to section 3.1 on line 177: "Different species may exhibit different C and N elemental and isotopic values under the same environmental and growth conditions (Mele et al., 2023), and within a single species, varying environmental conditions can lead to difference in these values (e.g. Magni et al., 2012; Briant et al., 2018).

**Line 207-208** It is not unsurprising that the lowest carbon isotopes are in those environments close to riverine input where the DIC would be influenced by the freshwater inputs reducing by ~-5 per mil. See studies examining environmental change e.g. Fitzer et al., (2019).

The above points need expansion into the discussion (~**lines 240**) of the importance of coastal environmental change for C/ N and what this tells us about the algal availability. This directly influences the C/N uptake into the tissues form environment and food source. How is this likely to change under climate change? What does this mean for the food web and availability of food for mussels and oysters in these coastal regions? This is key to support statements in the abstract relating to food web baseline data and management in a changing world.

We agree with the reviewer. We will add the following sentences on line 243: '$\delta$13C values of dissolved inorganic carbon (DIC) are lower in riverine/freshwater ecosystems compared to marine environments (Mook and Rozanski, 2000), resulting in more negative $\delta$13C values in phytoplankton, which may be ingested locally by bivalves. Additionally, $\delta$13C of particulate organic carbon (POC) from terrestrial sources tends to be more negative (Liénart et al., 2017) and can be consumed by bivalves. However, whether terrestrial POC is a significant food source for bivalves remains debated (Malet et al., 2008; Marín Leal et al., 2008; Dubois et al., 2014). Similarly, the $\delta$15N signal of freshwater and marine phytoplankton differs due to the distinct nitrogen sources in these two environments. Therefore, it is not surprising that the $\delta$15N signal in bivalves varies along a gradient related to proximity to river mouths.

**Lines 253-255** Figure 3 nicely presents the downwards shift in $\delta$13C at all sites, noticeably in locations near the Seine plume. Can the authors comment on changes to precipitation in the catchment of the Seine historically and how this relates to the downward shift in carbon isotopes? I.e. climate change, this could be expanded to support the findings and statements of the conclusions.

Based on existing data (Hydroportail ; EauFrance), over the period 1981-2021, there is no significant increase or decrease in the Seine river flows. As mentioned above, in depth discussion the effects of environment on bivalves elemental and isotope signal is not the object of this data paper. Bivalve elemental and isotope data, along with additional environmental data, will be thoroughly discussed in a forthcoming publication that will include environmental databases and examine the effects of global climate and regional factors (like river inputs and food availability), as well as the potential effects of bivalve species and physiology on the isotope signal.

**Response to R2 comments**

Lienart et al. present a highly interesting dataset on multidecadal carbon and nitrogen isotope values and the elemental composition of bivalves from the French coasts. The carbon isotope data have already been published (Lienart et al., 2024, Limnology and Oceanography Letters) and are accessible in a repository. The complete dataset is also available in another repository (Seanoe).

I find the manuscript suitable to support the publication of this dataset, which is unique, highly useful, and of exceptional quality. I have only minor comments, which I hope will be helpful in improving the manuscript :

We thank the reviewer for the valuable comments and suggestions, which have helped to improve the quality of our article.

**Lines 35–37**: Could you provide some examples of such indicators, please?

We modified the sentence on lines 35-37, which now includes examples of integrative and sensitive ecological indicators. "Yet, only a few ecological indicators are both integrative (i.e. measured in long-living species tissues) and sensitive enough (e.g. measured at molecular level such as metallothionein used as a biomarker of metal exposure, Amiard et al., 2006) to various disturbances while also exhibiting predictable responses with a low variability in its response (Dale and Beyeler, 2001; Niemi and McDonald, 2004)."

Amiard, J.C., Amiard-Triquet, C., Barka, S., Pellerin, J., & Rainbow, P.S. (2006). Metallothioneins in aquatic invertebrates: Their role in metal detoxification and biomonitoring. Aquatic Toxicology, 76(2), 160-202. https://doi.org/10.1016/j.aquatox.2005.08.015

**Line 54**: Bivalves can exhibit high carbohydrate values in the form of glycogen. Carbon is also a component of protein, with a C/N ratio of approximately 3.5. What is particularly indicative is how C/N deviates from this baseline value. Could you refine this section for clarity and precision?

We refined this section as following :

The C:N ratio is mostly an indicator of bivalve condition and physiology, reflecting the balance between organisms' requirements and elemental availability in the environment (i.e., ecological stoichiometry sensu Elser et al., 2003; N content increases (thus C/N ratio decreases) as protein content increases whereas C content increases (thus C/N ratio increases) as lipid or carbohydrate content increases).

**Lines 63–64**: I appreciate the idea of using these values in models. Could you expand on this concept in Section 5? Additionally, consider mentioning this in the abstract.

We slightly modified the sentence lines 63-64 and added the following sentence about using such data in models in Section 5.

"Ultimately, it could provide valuable input for developing predictive models of **bivalve physiology (Emmery et al., 2011) or trophic ecology (Marín Leal et al., 2008) explaining** ecosystem response to future environmental changes and possibly forecast potential impacts of climate change and human activities on coastal ecosystems."

Emmery, A., Lefebvre, S., Alunno-Bruscia, M., & Kooijman, S. A. L. M. (2011). Understanding the dynamics of δ13C and δ15N in soft tissues of the bivalve Crassostrea gigas facing environmental fluctuations in the context of Dynamic Energy Budgets (DEB). Journal of sea research, 66(4), 361-371. https://doi.org/10.1016/j.seares.2011.08.002.

Marín Leal, J. C., Dubois, S., Orvain, F., Galois, R., Blin, J. L., Ropert, M., ... & Lefebvre, S. (2008). Stable isotopes (δ13C, δ15 N) and modelling as tools to estimate the trophic ecology of cultivated oysters in two contrasting environments. Marine Biology, 153, 673-688. http://dx.doi.org/10.1007/s00227-007-0841-7

**Lines 84–85**: Are there any recordings of individual size other than the given ranges? Including such information would have been interesting.

We agree that this information would have been interesting. Unfortunately, there is no such information in the ROCCH database. Note that all the individuals are pooled in a single meta-sample (see below). Thus, the only usable value would be the mean size.

**Line 106**: The archived samples are already pooled, correct? Could you provide more details about the grinding procedure? If samples were already ground initially (line 91), additional grinding might be necessary. Sample homogeneity is crucial for isotope analyses.

We agree that sample homogeneity is crucial for stable isotope analysis and we were careful about it when analyzing the archived sampled. Archived sampled were already pooled, grounded and homogenized before long-term storage as described in section 2.1. lines 85-94. We modified a few words in section 2.1. for more clarity.

**Line 110**: Could you provide the values for the standards used? For instance, the caffeine standard from IAEA (IAEA-600) is not certified for nitrogen analysis ("Reference material IAEA-600 is aimed at δ13C calibrations by TC/EA technique; its δ15N is provided for informational purposes"). Could you discuss this in the manuscript?

Indeed, caffeine is not certified for nitrogen analysis, but calibration was performed with internal standards. We added information about the standard values and calibration on lines 110-114 :

- Casein (δ13C -23.30 ‰; δ15N 6.30 ‰)
- Glycine (δ13C -45.20 ‰; δ15N 3.00 ‰)
- Caffeine (IAEA600: δ13C -27.77 ‰, δ15N 1.00 ‰) not certified for N
- Graphite (USGS24: δ13C -16.05 ‰)

**Line 161**: It is unclear whether the slopes were derived from this package. Usually, the Sen slope is used, and this seems to be the case. Could you clarify and specify the function used from this package?

We used linear models to calculate the slopes. We added missing packages and function names on lines in section 2.4 and the corresponding references to the manuscript.

**Figure 2 and text**: I would like to see the patterns of C% and N% separately. It is important to examine how C and N evolve independently. The dataset appears to report elemental composition in g/g, while the C/N ratio is expressed in mol/mol. Please specify this explicitly.

We modified Figure 2 and add the C% and N% data and specified the units in section 2.3.

**Line 229**: I believe the authors are referring to the C/N ratio here, but the trends for C and N individually are also valuable.

We agree that C and N% are also valuable and we included C and N% data description in the entire manuscript.

**Section 3.3.1**: Use a capital "P" (also in Sections 3.3.2 and 3.3.3).

It was an oversight. We corrected all section's titles.

**Lines 289–290**: I agree with this suggestion—please include these visualizations and analyzes (as noted in my comment above).

We added the visualization of C and N% in figures 2 and 3 and included its description in the ms.

**Figure 5**: Additional figures showing patterns for C and N would be helpful.

We added C and N % data in Figure 2 and as a new Figure 6 (similar to Figures 3, 4 and 5).

---

## Author Response (AR2)

Camilla Liénart
camilla.lienart@u-bordeaux.fr
(+33) 6 72 57 81 61
Université de Bordeaux
UMR 5805 EPOC
Station marine d'Arcachon
2 rue du Professeur Jolyet
33120 Arcachon
France

Arcachon, January 6th, 2025

Dear Editor,

We are pleased to submit the corrected version of our manuscript entitled "Bivalve monitoring over French coasts: multi-decadal records of carbon and nitrogen elemental and isotopic ratios as ecological indicators of global change", by Camilla Liénart, Alan Fournioux, Andrius Garbaras, Hugues Blanchet, Nicolas Briant, Stanislas F. Dubois, Aline Gangnery, Anne Grouhel Pellouin, Pauline Le Monier, Arnaud Lheureux, Xavier de Montaudouin and Nicolas Savoye for a submission in *Earth System Science Data*.

We thank the reviewers and editor for their valuable comments and suggestions which have helped improve the quality of our article. We took into account all comments from the reviewers 1 and 2, as well as the suggestions from the editor. We hope that our manuscript is now acceptable for publication in ESSD.

Sincerely,

Camilla Liénart, on behalf of all co-authors